# Nothing in Excess: Physical Activity, Health, and Life World in Senegalese Fulani Male Pastoralists, a Mixed Method Approach

**DOI:** 10.3390/ijerph20216999

**Published:** 2023-10-30

**Authors:** Dominique Chevé, Enguerran Macia, Moussa Diallo, Loic Lalys, Amadou Hamath Diallo, Sidaty Sow, Audrey Bergouignan, Priscilla Duboz

**Affiliations:** 1IRL 3189 ESS, Faculty of Medicine, Cheikh Anta Diop University, Dakar BP 5005, Senegalenguerran.macia@cnrs.fr (E.M.);; 2Department of Modern Literature, UFR LASHU, Assane Seck University, Ziguinchor BP 253, Senegal; moussadiallosn@yahoo.fr; 3UMR 8045 BABEL, Institut Médico Légal, 2 Voie Mazas, 75012 Paris, France; 4UMR 7178, IPHC, 23 Rue Becquerel, CEDEX, 67087 Strasbourg, France

**Keywords:** biological anthropology, Africa, great green wall

## Abstract

Objectives: The goal of this study was to evaluate the application of the short form of the International Physical Activity Questionnaire Survey (IPAQ-SF) in the rural Senegalese Fulani pastoralist population by combining quantitative and qualitative methods. Design and participants: For the quantitative method, 101 men completed the IPAQ-SF questionnaire measuring moderate, vigorous, and walking physical activity. Self-rated health, BMI, and sociodemographic variables were also collected. With regard to the qualitative methods, a total of 22 participants were recruited and interviewed. Four themes were addressed, including (i) physical activity (PA) and its definition, description, related experiences, and representations of social actors; (ii) PA and health; (iii) PA and sport; and (iv) the body and Fulani world of life (i.e., Pulaagu/Ndimaagu). Results: Sahelian herders have a high level of self-reported PA and a low amount of daily sitting time. The measure of PA as proposed by the IPAQ-SF is not adapted to the Senegalese Ferlo pastoralists, mainly because this scale gives too much importance to leisure-time PA, perceived as unproductive energy expenditure, which is factually and symbolically antinomic to the Fulani lifeworld. Thus, neither intense nor moderate PA is related to self-rated health. However, sedentary lifestyles are linked to self-rated health and, therefore, to mortality and morbidity in Fulani pastoralists. Finally, walking, which is the dominant PA during transhumance and herd surveillance, is related to BMI. It therefore represents a protective factor against the occurrence of overweight and associated chronic non-communicable diseases. Conclusion: The mixed method approach developed in this study has shown that the IPAQ-SF is not a valid measure of PA in the population of Fulani male herders from the Ferlo region, given that unproductive energy expenditure is incompatible with the Fulani way of life, which condemns excess and immoderation.

## 1. Introduction

The role of physical activity in human health and disease is well established [1,2]. Physical activity is now prescribed in some countries (e.g., France) as a tool for managing chronic diseases [3]. Globally, 7.2% and 7.6% of all-cause and cardiovascular disease deaths, respectively, are attributable to physical inactivity (i.e., not reaching currently recommended levels of physical activity) [4]. Numerous scales or surveys, including the International Physical Activity Questionnaire, or IPAQ, have been developed over the years to quantify habitual physical activity within and across different populations [5]. The IPAQ-SF has specifications for respondents aged 15 and over: predictive validity, concurrent validity, convergent validity, criterion validity, and discriminant validity [6]. These are appropriate research tools, and a good test-retest is an indication of reliability [5,7]. However, most research on the relationship between physical activity and health outcomes has been conducted in high-income countries [8,9,10]. Findings may not apply to low- and middle-income countries given that physical activities and the form they take are highly dependent on the socio-cultural context in which they are carried out. It is essential to validate tools for measuring daily physical activity in low- and middle-income countries, given the considerable increase in the prevalence of chronic diseases in these countries [11], in addition to the high mortality rate linked to non-communicable diseases [12]. Moreover, the relationship between the increase in chronic diseases and the decrease in physical activity [13] makes it all the more necessary to address this gap in knowledge. The goal of this study was to test the application of the short form of the IPAQ survey (IPAQ-SF) in the rural Senegalese Fulani pastoralist population by combining quantitative and qualitative methods [14,15,16,17]. Furthermore, we have attempted to define the dimensions of physical activity that the IPAQ-SF captures in this isolated rural environment and, more importantly, what it means to be physically active in this population of Ferlo Fulani men. This research has scientific and clinical implications by helping to better understand the lifeworld of the Fulani herders of the Ferlo and their relationship to physical activity, the body, and health.

### 1.1. Theoretical Framework

Physical activities involve the body, a bio-cultural object per excellence, the use of the body, as well as the bodily practices that are constructed by their permanent and dynamic interactions with the physical and social environment [18]. No body movement, physical activity, or health status is the sole result of biological predisposition or socio-cultural factors but rather of their combination. The bodies and their determinants are always embedded in dynamic socio-cultural constructs, both practices and uses, historically constructed ways of life, and environments that are both diverse and specific but porous. They are thus influenced by complex logic, shared imagination, axiological determinations, transitions of practices, and diverse hybridizations [19].

Understanding these complex relationships and interactions between physical activity, the body, health outcomes, and socio-cultural and physical environmental factors requires a mixed-methods approach combining quantitative and qualitative methods. This is the approach we propose to employ to examine physical activity in relation to the pastoral lifestyle characteristic of the Fulani culture in the Ferlo, located in Northeastern Senegal. Such an approach will allow for a more holistic understanding of the socio-cultural determinants of physical activity, the nature of these activities, their characteristics, and the mutations in progress in Fulani society due to the ongoing urbanization.

The notion of “lifeworld” [20,21] allows us to target the complexity of the environmental and social universes of a human group. This includes the value system, norms, and cultural particularities, along with their collective history. These worlds of life are porous and diachronically oriented but respond to complex logic. The Fulani’s lifestyle evolves around livestock, which shapes the Fulani’s world of life. Their world of life includes livestock and harsh and sober living conditions.

### 1.2. Context

The epidemiological transition in Sub-Saharan Africa began several years ago and is ongoing in Senegal. This transition, along with demographic and nutritional transitions, is one of the major determinants of the increased burden of non-communicable diseases [22,23]. Much of the primary literature on epidemiological transition and physical activity is based on studies conducted in economically advanced countries or among urban sedentary populations in Africa [24]. Although a few articles focus on physical activity practices in sub-Saharan African rural areas [25,26], there is still a lack of data on physically active populations in low- and middle-income countries. This is why it seems necessary to describe the socio-economic, demographic, and health situation of the Fulani population in the Ferlo region of Senegal.

The Ferlo sylvo-pastoral reserve, located in Northeastern Senegal, is rural and relatively isolated. It is inhabited mainly by Fulani populations, who are essentially pastoralists and undertake transhumance (i.e., leaving for several months in search of pasture to accompany the herds) [27] during the dry season. Men’s roles are traditionally associated with herding, while women look after small ruminants and domestic tasks in the camp or villages [28]. In this Sahelian environment, living conditions are harsh: access to water and basic services (health, electricity, transportation, etc.) is very limited [29]. The commune is structured around three villages: Widou-Thiengoly, Amaly, and Téssékéré, each with a borehole. About half of the population lives in these villages; the other half lives in camps spread between the villages, located within a 15-km radius. This savanna is shrubby to woody, the result of ongoing environmental degradation since the 1970s [29]. In this traditionally pastoral population, the relationship of the Fulani to livestock goes beyond the mere necessity of their subsistence through professional activity. It is a kind of identity imperative: to be a Fulani man in the Ferlo is to be a shepherd (herder or shepherd), not occasionally or by accident but almost necessarily, ontologically.

The Fulani world of life in the Ferlo is particularly linked to the constraining conditions of existence. These conditions determine the lifestyles and activities that are gendered and distinctive across age groups. They also determine behavioral and social interactions. At the axiological and symbolic level in this society, the behavioral appreciations (ndimaagu) and the values of the pulaagu permanently set a model for rectifying what needs to be rectified [30,31,32,33]. The pulaagu (or Pulaaku) is the code of prescriptions and proscriptions of the Fulani community, not as a Fulani invariant that would freeze the population in an identitarian interpretation, but as shared and recognized norms, values, and imaginaries. Ndimaagu comes from Dimo, which is calm, serene, not demanding in any way, and satisfied with the little he has, even in iniquity. In this sense, every free man must control himself in all things and circumstances and not let himself go to excess. Sobriety (yoneede) is required in the Fulani world of life. Pulaagu touches on all codes—dress, food, etc.—related to one’s gait, the way one speaks, how to treat strangers, how to avoid an unseemly situation so as not to embarrass one’s neighbor, etc. This is the case, for example, of discussions regarding a physical trait or an attitude that contrasts with those, supposedly fixed and valued, of the Fulani of the Ferlo; or again, with regard to morality, a certain number of behaviors such as gluttony, fear, and the company of women without a reason that is acceptable according to these codes are denounced, repressed, suppressed, or mocked. These normative “dispositifs” construct “any good Fulani”, without selection of gender or age; it is a question of being the right man or woman.

The modes of regulation and maintenance of balance in this Fulani world of life in the Ferlo are quite coercive through the imposition of these norms and values. The one in the group who does not conform sufficiently, who does not fully embrace this way of life, is said to be “o sonngaaki” (from the verb “sonngaade" in the negative form, to say that the person in question is not listening enough. To be more accurate, sonngaaki is used to describe someone who does not take hints in a cultural context where everyone is smart enough to pick up on the unspoken. This negative judgment can be harder to bear than a beating with a stick or a knife), a designation that can be fearsome. Here, as elsewhere in West Africa, whether urban or rural, reputation is of utmost importance for social existence [34], and not fitting in, being the object of this disapproval because one does not understand the injunctions to conform, is stigmatizing.

### 1.3. Research Object, Problem and Issue

In the Fulani way of life, physical activity essentially corresponds to walking, which is linked to the livestock-rearing method. Transhumance determines the annual rhythm. The installation of boreholes and the relative socio-economic development of the region have recently led to a reduction in transhumance, both in terms of duration and distance [35,36]. Yet walking remains fundamental and necessary, as motorized vehicles are almost absent from the area. Even if herders are currently diversifying their income (through real estate, trade, and transport), livestock breeding remains the cornerstone of the Fulani people’s way of life [37]. Thus, for both cultural and material reasons, recurrent and relatively intense daily physical activity is characteristic of the Fulani populations of the Ferlo, particularly among men, who are responsible for grazing the herds. Furthermore, this high level of habitual physical activity is observed in a population with a low prevalence of overweight and obesity [38]. This rural Fulani population should therefore appear as an ideal test of the relationship between physical activity and health [39], since the level of physical activity in this population could be above a certain threshold for which excessive weight gain and associated metabolic risks would be observed [40]. However, this issue is not purely physiological, and the anthropobio-cultural characteristics of populations also determine body use and energy expenditure [20,41]. In this general context, we proposed to examine associations between the IPAQ-SF-derived physical activity outcomes and health outcomes in this population of Senegalese male pastoralists (i.e., health, BMI), while taking into account what physical activity means for the populations.

## 2. Material and Methods

### 2.1. Quantitative Approach

#### 2.1.1. Sampling

Cluster sampling was used. Out of a total of 20 villages, after the first random sampling, ten primary clusters of villages were selected. Afterward, a secondary random sampling within each village allowed for the selection of houses. The choice of houses was made according to the survey method of the World Health Organization (WHO), in which we chose the houses by moving closer and closer to the house first selected randomly. Once the house was chosen, all the people living in it who met the selection criteria (aged 15 or over, living in the commune of Téssékéré, and not physically disabled) were included in this study. A maximum of 15 people was set by village, making a total of 150 participants for the ten selected villages. They were then called to the health center to complete biometric measurements and a socio-demographic interview. Out of the 150 men selected, 101 reported to the health center and were included in the study. Two investigators completed standardized training to avoid classification bias. The language of the interviewers and respondents was Fulani. The questionnaire had previously been translated from French into Fulani using the parallel inverted translation method [42]. Recruitment took place every day for the last two weeks of May 2021.

#### 2.1.2. Outcomes

##### Self-Reported Physical Activity

Components of PA contents were calculated using the following formula:-moderate PA = 4 METs (metabolic equivalent) × average minutes of daily moderate physical activity × number of days with moderate PA-vigorous PA: = 8 METs × average minutes of daily intense physical activity × number of days with intense PA-walking: = 3.3 METs × minutes of walking activity × number of days with walking bouts-Total METs = sum of the 3 METs (intense, moderate, walking).

MET data were then converted using the rule that MET < 600 MET·min/week is in the low category, 600 ≤ MET < 3000 is in the medium category, and MET ≥ 3000 is in the high category (Guidelines for Data Processing and Analysis of the International Physical Activity Questionnaire (IPAQ)-Short Form, 2004).

Furthermore, sitting time was assessed using the following IPAQ-SF question: “During the last 7 days, how much time did you spend sitting on a weekday (hour and/or min per day)”?

Two PA-related questions were also included in the questionnaire:-Are your physical activities always related to your work? (Yes, No, Sometimes)-Are your physical activities keeping you healthy? (Yes/No)

##### Self-Rated Health

Self-rated health was measured using a questionnaire with five possible answers: “Overall, would you say that your health is excellent, very good, good, fair, or poor?” For the majority of bivariate and multivariate analyses, this variable was dichotomized. In accordance with Jylhä’s reflection [43] showing a break between good health—“the baseline that does not normally need to have a cause”—and less than good health, the split was made between the answers “excellent”, “very good”, and “good” (scored 0) and the answers “fair” and “poor” (scored 1).

##### Body Mass Index

Following WHO recommendations, BMI was calculated by dividing weight (kg) by the square of height (m^2^). Weight was measured using a digital scale (measurement accuracy of 100 g), with subjects dressed in minimum clothing and barefoot. To measure height, the subject stood with arms at the sides and heels joined. Thinness was defined as BMI < 18.5 kg/m^2^; normal weight as 18.5 kg/m^2^ ≤ BMI < 25 kg/m^2^; overweight as 25 kg/m^2^ ≤ BMI < 30 kg/m^2^; whereas obesity corresponded to a BMI of ≥30 kg/m^2^. For the sake of analysis, people with excess weight (BMI ≥ 25 kg/m^2^) were distinguished from others (BMI < 25 kg/m^2^).

##### Socio-Demographic Variables

Among the socio-demographic data collected during the interviews, three were taken into account in this study: age (< or ≥ to 40 years old), type of occupation, and marital status (single/divorced/married/widowed).

##### Statistical Analysis

Bivariate and multivariate analyses were used to test associations between dependent and independent variables. Bivariate analyses included Chi^2^ tests and Student’s *t* tests for mean comparisons. Logistic regression was also used to assess the extent to which age and sitting time predicted poor self-rated health. The software used for the statistical analysis was IBM SPSS Statistics V. 22. Results were considered significant when *p* < 0.05.

### 2.2. Qualitative Approach

#### Recruiting

The qualitative phase of this study was conducted in parallel with the quantitative data collection. Qualitative data was collected using semi-structured interviews. Physically active adult men (pastoralists, shepherds who may have other jobs or functions within the community, such as traders, gardeners, local agents, etc.) of 30 years old and over were recruited. Particular attention was paid to balancing representation between people living in the village (i.e., near the borehole, the market, or the health center) and those living in camps in remote areas. A total of 22 study participants were recruited and interviewed. Four themes were addressed, including (i) PA and its definition, description, related experiences, and representations of social actors; (ii) PA and health; (iii) PA and sport; and (iv) the body and Pulaagu/Ndimaagu. All participants were informed of the voluntary and free nature of their participation. All the interviews were conducted in Pulaar, then translated and transcribed in French in their entirety with the agreement of the participants. This was done thanks to the prior development of a lexicon corresponding to vernacular usage, which allowed us to identify fine nuances and work on the accuracy of the terms. Table 1 presents the distribution by age and location of the study participants.

As recommended by Bender and Ewbank [44], we developed a set of questions to guide and direct the discussion beforehand. The interviews were conducted in Fulani and lasted one hour and 15 min on average, but ranged from 45 min up to two hours.

### 2.3. Data Analysis

Data analysis was conducted using the qualitative thematic analysis method described by Mason [45]. The first phase of this analytical method concerns the identification of themes. The second phase is interpretative and conceptual. The thematic analysis consists of several stages. First, the themes were identified by reading the interviews several times. DC, PD, and EM worked independently to identify and name these dimensions. After discussion among the authors, agreement was reached on the dimensions. The quotations that best illustrated these dimensions and their causal relationships were then selected. Conceptual analysis, on the other hand, is considered more subjective than thematic analysis [46] and consists of an interpretation of discourses. 

### 2.4. Ethical Concerns

Ethical approval was provided by the Comité National d’Ethique pour la Recherche en Santé (Protocole SEN 13/67).

## 3. Results and Discussion

### 3.1. Structure and Characteristics of the Population Sample

As shown in Table 2, the 101 male study participants were, on average, 43 years old. The vast majority of individuals were herders or pastoralists (71.29% in total), and all of them were married. None of them were single, widowhood being absent due to the practice of polygamy. Overall, this population of men had a high level of self-reported PA that was comparable to the one found in Nigeria [8]. As expected, this population had a relatively low mean BMI (WHO data on mean BMI, https://www.who.int/data/gho/data/indicators/indicator-details/GHO/mean-bmi-(kg-m-)-(age-standardized-estimate) accessed on 6 may 2023) and a low amount of daily sitting time, especially compared to European populations (20 min/day in our sample vs. 25 in France and 25.6 in both Italy and Portugal) [47,48].

### 3.2. No Unproductive Physical Activity

In the IPAQ applied to European populations, recreational sport is largely included as a preponderant factor for physical activity. By contrast, in the Ferlo, if sport is practiced by the youngest, notably at school or sometimes during soccer games organized at the water and forestry base (Widou), it remains decried and negatively judged:

“*It is not part of our culture to go running and make movements like I see young people doing it here now*”.(Herder, E7)

Nobody can understand and admit that a male adult can engage in sport; he would be considered a fool by the community: 

“*Whoever practices sport is considered mentally retarded or a scoundrel”. When they see me, for example, running, people will say that I am mentally ill and that I have nothing to do. “Instead of being there running, I should go and take care of my herds, because there are many things to do there*” (Herder, E22). “*But [an adult person] cannot go running; imagine an old man on a road running; that’s catastrophic”. They’re going to think he’s just gone crazy (laughs)*”. (Herder, E9).

In short, the leisure sports practiced in the cities cannot be encouraged here because they are unproductive:

“*In general, if you meet a Fulani woman, she will ask you why you are running (laughter). If you meet a man, the first thing he will say is that you have no activity to do; you have too much free time*” (Herder, E1). *Furthermore, “some people can say that one runs all the time; he has too much free time to waste*” (Herder, E4). “*As I said earlier, our sport is walking behind our herds of cattle; on the other hand, the other sport [sport as leisure, unproductive] is people who have nothing to do who do that (laughs); if we have something to do, we’re not going to be there running around in the middle of nowhere even if it’s good for our health. Anyway, there are always activities that can replace sports*”. (Herder, E21).

This expenditure of energy is considered to be wasted and, therefore, without any gain or profit for the farm or the community, and cannot bring any benefit. Far from the report to the enjoyment of power through prolix expenditure that Georges Bataille analyzes in the potlatch or kula, following Boas [49] and Mauss [50,51], of the “squanderer” donors, whose prestige thus acquired is a symbolic and real gain,

On the contrary, the pastoralists of the Ferlo see this expenditure of energy, this sport, only as a Western and/or urban deviation and, at best, as a possible activity for young people within the framework of school or possibly very occasionally and therefore in a contingent manner:

“*Currently, we see more and more young people practicing sports; it is only the young people that we see here running in the streets*” (Herder, E9), *or “the young people do sports like soccer and the adults do not do sports, but they are the young students of the different villages*” (Herder, E12). One of our respondents summarized it this way: *“Here, in the area, there are only young people who do sports, especially young people who come from elsewhere in the cities, but not the young people of the village who are interested in the herds of cattle*”.

The energy expenditure linked to leisure physical activity, which is unproductive, is therefore excluded and does not make any sense to them. The quantitative approach confirmed this representation. Indeed, for 80.2% of the men interviewed (i.e., 81 individuals), physical activities were only related to work. However, the IPAQ, developed in Western countries, includes an essential “leisure” component. In the Ferlo, however, all efforts are directed towards work, social usefulness, breeding, monitoring, and/or caring for the herd.

It is clear that sport was only considered “good for health” by a minority of respondents and that none of them practiced it for health reasons:

“*Of course, there is a link between sport and health: if you do sport, your blood circulates normally and your body is in good shape; that is health*”.(Herder, E9)

Most of the participants (80.2%) considered that physical activities did not keep them healthy. On the contrary, they linked a potential deterioration of health to excessive physical activity, because of the fatigue caused and the obligation to walk mainly to look after the herd. Moreover, in our interviews, the verb “to be forced” or the term “constraint” came up repeatedly and systematically. Physical activity was inexorably and essentially linked to the work of the shepherds; they were obliged and constrained to do it.

They know and have integrated the norms of global health conveyed in Western prescriptive discourses, which value sport and health [52,53]. For example, one of our respondents (E9) declared:

“*Any activity that we do, if this activity is not intense, can allow the body to be in good shape, and the blood will circulate well. We can consider this activity a sport. On the other hand, if the activity is intense, it can lead to health problems”.*


They did not deny the benefits of sport because they knew that health professionals talked about it, but they considered that doing sport did not make sense in the Ferlo. This sporting activity is irrelevant. It is only good for urban people or those who have time. Their constant walking is enough to keep them “in good shape”:

“*For me, in the sport you are talking about here, only people who do not make too much physical effort are forced to do that to help their body sweat a little in order to evacuate the sugar or salt content of their body. A herder, on the other hand, doesn’t need to do that because he is always under the sun, walking all the time*”.(Herder, E20)

Even if perceptions have been gradually changing and if today it is no longer a drama for old people to see a young herder jogging from time to time.

Pastoral activities are carried out in difficult and precarious living conditions. Therefore, all the physical effort and expenditure of energy, as well as the resulting fatigue, can only be justified by work. They cannot fall under leisure. 

“*The people I see doing sports here are the teachers and the young pupils and students; but I have never seen a grown-up man living in the surrounding villages doing sports here*”.(Herder, E8)

### 3.3. Walking as an Adequacy Condition for Identity

Sport is a leisure activity; walking is a necessity. There cannot be any unproductive energy expenditure. All efforts must be efficient and linked to the physical activities necessary for breeding. This statement, which summarizes the adequacy between physical activities and breeding—work, predominated in our interviews:

“*All the physical activities that I do are related to breeding; I do not do any other activities*”.

Indeed, all the activities address this need.

The IPAQ-SF-derived physical activity was high in the Fulani male population. This was likely due to a lifestyle essentially oriented towards energy expenditure dedicated to the performance of permanent physical work (walking, drawing water from the well, lifting heavy loads, etc.):

“*There are also certain work-related activities that make the body move apart from walking; there is also the fact of fetching water from the borehole, or going with the herds into the forest, or the act of fetching water from wells*”.

Daily practices related to work and subsistence, in this case essentially walking, determine the construction of bodies and their uses [41] and identity [18,19,54].

As expected, total MET was negatively related to participants’ BMI (Pearson: −0.249; *p* = 0.033), which is consistent with the literature [47]. Interestingly, the relationship between time spent walking and BMI was even stronger (Pearson = −0.303; *p* = 0.002) (Table 3). This seems to be in agreement with the qualitative approach developed in this work.

Walking is the main physical activity of the Fulanis and a major component of their pastoral identity. If it is not a question of “*running in the void because we have enough time to waste…*” (Herder, E21), a true Fulani (it must be noted that our study population constantly designates and defines itself according to a sort of paradigm of the “true Fulani”, of a “Fulani being” as a herder, and that this membership in a Fulani community is claimed) is always active, always walking:

“*A Fulani herder is obliged to be courageous, resistant, and very flexible to carry out his activities as a herder. A herder is obliged to move all the time because he is with the herds of cattle, and the herds cannot stay in one place; they have to move to go and look for pasture*”.(Herder, E14)

The heads of households, their sons in charge of the herds, and the herders to whom the daily management of the herd has been delegated [55]; they all walk. Even if lifestyles (appearance of the use of motorcycles and sedentarization; 22) have changed, walking, for both men and women, is consubstantial with the Fulani body and pastoral tasks.

It is therefore an essential element of energy expenditure in Fulani society, and by extension, in many rural sub-Saharan African societies:

“*For me, physical activity means walking beside my herds*”.(Herder, E8)

This physical activity concretizes the adequacy of the Fulani man and his herd. The pastoral ideology has often been posited as the identity basis of the Fulani, regardless of the African territories [56].

If the herd goes, everything goes, one could say, and the efforts related to long and regular walks are well supported:

“*The Fulani is the language first, then the most important activity is pastoralism; the Fulani must be a herder who lives with the herds in the bush. Now, [Finally] come the other activities”. “To sum up, a true Fulani is one who has herds and who lives with and for his herds*”.(Herder, E22)

In short, fitting in with his herd in good shape makes the herder himself “on the move” and “in shape”. This state refers to an often-evoked disposition of contentment and well-being [57]:

“*If we are talking more exactly about the breeder, his energy depends on the state of his herds of cattle”. If he sees his herds of cows, for example, well satiated with plenty of calves and milk, He is very happy. “Consequently, he has energy*”.(Herder, E14)

When asked what “having energy” means, one rancher (E10) responded:

“*A rancher who does not have enough grazing problems and whose herds are not starving, they are well satiated, will necessarily have the joy of living, therefore he has energy […] because he is joyful to see his herds in good shape*”.

This equivalence state of good shape, well-being, and joy in correspondence with the state of their herds determines the experience of walking. In fact, this state of “joy” (“hakkille deeydo”, peaceful spirit) overcomes the necessary and sometimes exhausting efforts of long walks.

In this respect, many of our respondents stated that the “good Fulani”, the “true Fulani”, is always walking, moving, and active, and this is a sign of balance and good health. The more one sits, the less healthy one is, regardless of age. The correspondence between the need to be active and being in good health is therefore obvious. This is particularly evident in the quantitative analysis developed. Table 4 shows that, regardless of age, sitting is significantly associated with poorer self-assessment of health, which is in line with previous similar studies [58].

Thus, even an elderly man will not remain physically inactive for too long, otherwise, his health will deteriorate.

### 3.4. Physical Activity: Right Dose and Health

But this physical activity must not be too intense, too violent, too tiring, or too unbalanced in duration. There must be moderation. Calculating and controlling efforts are necessary. Moreover, the way in which health is assessed in the Fulani pastoralist population is not related to physical activity (*t* = 0.045; *p* = 0.964).

In the Fulani population of the Ferlo, to maintain good health, nothing should be overdone. 

“*These activities make us weak. When it is too much, because sometimes we go weeks without sleeping or eating during transhumance, it can even kill us or cause illness*”.(Herder, E7)

Temperance is needed in everything. Moreover, excess (“furßeende”) is prohibited. Moderation is said to be “deeyre” and corresponds to restraint, to “ndimaagu”, that is to say, controlled behavior, calm and sobriety, even serenity, and balance. This perfect equilibrium is achieved when conflicting forces and a safe distance are mastered in any activity or situation.

It is clear that the declared “good health” is linked to the adequacy between the satisfaction of being able to meet daily needs and the necessities of breeding and a reasonable capacity, which is notably translated by a rejection of excess in all things. This conception of health, as well as of the good life, refers to the notion of good balance in all activities, those related to pastoral tasks as well as those related to the gestures of daily life, behaviors, and, more broadly, the Fulani way of life. Even rest must be moderated:

“*Resting is good, but too much rest is bad for the Fulani”. “The Fulani do not like somebody who has too many hours of rest*”.(Herder, E7)

Indolence is seen as an excess of complacency and a lack of resistance: 

“*A Fulani who does nothing is insignificant”. “He is not considered at all here; to have value in the eyes of others, you have to be a good worker; people start to give your example to others; you have not seen the son of such and such how brave he is, etc*”.(Herder, E22)

Thus, for example, sobriety (yoneede) as well as reserve are valued in all areas of life. It is a question of controlling oneself and resisting any excess, whether it be food or work, desire, activity, rest, or words. A typical ideal emerges in the statements, both aesthetically and ontologically: 

“*A true Fulani is neither too black nor too light;, he is just in the middle; the Fulani breeder is, in general, thin, slender but very pretty, very active, very lively, and very flexible*”.

Thus, one of our respondents (Herder, E8) said: 

“*In relation to the body of a good herder, we prefer a flexible Fulani who is very active and very resistant to any activity”. In general, we do not like people who are too fat and have difficulty moving. These types of people are a burden for the family because they do almost nothing. What is good for a Fulani is to be able to control his body*”.

These comments by our respondents invariably referred to the behavioral norms of the ndimaagu, which seem to echo the ancient Greek Epicurean and Stoic conception of temperance and moderation, of the right dose or just dose of everything [59], whether it is a question of the appreciation of affections (i.e., that which affects us, both on a sensitive and physical level, in this case, for our respondents, not resting too much, not eating too much, not sweating too much, not getting too agitated, etc.) or acts (not speaking too much, neither too often nor too loudly, not working too much, etc.). This may shed light on the Fulani’s ideals of behavior, aesthetics, and ontology.

The prescriptions of self-control and body control exclude “too much” in all things. In an extremely restrictive Sahelian environment, controlling one’s body is a social injunction that corresponds to the implacable harshness of the physical environment and the determining conditions of existence. It is a matter of resisting and continuing to exist in a physical and cultural environment that forces sobriety and a certain asceticism—nothing too much because the environment does not allow it. In fact, the representations produced by the Fulani give coherence to this living world. Everyone can recognize themselves in it and share these representations and imaginaries. In short, the physical, anthropic, and social environment, as well as its necessities, here as elsewhere, determine the symbolic productions. The life of the Fulani, rough and precarious, does not allow for a taste of abundance and unbridled consumption. Minimalism is the rule. The declarative construction of these social representations by our study population seems to us to be a kind of collective idiosyncrasy.

## 4. Conclusions

The general objective of this article was to evaluate what the IPAQ-SF actually measures in this population of Senegalese Ferlo pastoralists. We show that the measure of physical activity with the IPAQ-SF is not adapted to the Senegalese Ferlo pastoralists. This is mainly because this scale gives too much importance to physical activities that, as shown by the qualitative study, cannot exist in this population. Thus, neither intense nor moderate physical activity is related to self-rated health. However, sedentary lifestyles are linked to self-rated health and, therefore, to mortality and morbidity in Fulani pastoralists. Finally, walking, the dominant physical activity during transhumance and herd surveillance, is related to BMI and therefore represents a protective factor against the occurrence of overweight and associated chronic non-communicable diseases. Time spent walking, under these conditions and in this population, appears to be a much more precise etic indicator of the objective health of individuals than the generalist measure provided by the IPAQ-SF (bearing in mind that the IPAQ-LF is unsuitable because of the measurement of leisure-time physical activities). 

The paradoxical ambition of the international character of the Western-centric measurement system of physical activity, often questioned [8,9,10], is also invalid here. The living conditions of Fulani pastoralists create a particular physical activity profile, measurable by factors related to the physical and cultural environments in which they live. The qualitative approach developed in this work has made it possible to show that unproductive energy expenditure is factually antinomic (i.e., it is impossible because it would jeopardize the energy necessary for daily life) and symbolically antinomic (i.e., it has no meaning or reason to exist, is not justifiable, and is even reproved and mocked) to the Fulani lifeworld. This qualitative approach has also highlighted the mode of vital and social regulation that condemns excess and immoderation in all Fulani activities and behaviors. 

This study focused only on Fulani male pastoralists from the Senegalese Ferlo. Future studies should take females in this area into account in order to test the adequacy of the IPAQ for the physical activity of this population category. In addition, if the analysis conducted is valid in the sylvo-pastoral zone, it may not be valid in the same way in the Ferlo zone located further east, where pastoralists and farmers co-exist. Finally, the quantitative estimation of the pattern of physical activity should be complemented with objective measures of physical activity derived from 3D accelerometry, since a discrepancy can obviously be observed between the declarative measurement of physical activity (IPAQ SF) and its physiological measurement. This could be further combined with measurements of free-living total daily energy expenditure and activity-related energy expenditure measured by the gold standard method of doubly labeled water.

Social and family organization patterns are changing, and geographical conditions are changing, as are the means of communication (the introduction of smartphones, motorcycles, and various screens, still to a relatively limited extent). The system of adequate values based on living conditions remains prevalent, but the transformations caused by the opening up of the area are likely to weaken it by undermining the normative mechanisms and modifying behaviors, particularly those that determine the occurrence of chronic diseases (diet, physical activity, etc.). This is precisely what is being researched in the Senegalese Ferlo, in particular by the Téssékéré International Human-Environment Observatory (https://ohmi-tessekere.in2p3.fr/, accessed on 18 September 2023). These physical and cultural environmental changes will modify the lifeworld of the Senegalese Ferlo Fulani, and, with respect to our topic, perhaps then it is possible that IPAQ will become a relevant measure of the correlation between physical activity and health for this population.

### Strengths and Limitations of This Study

This study focuses on physical activity and its measurement in an isolated rural sub-Saharan African population.

Through the application of a mixed method, this study interrogates in a new way the adequacy between the IPAQ-SF scale and physical activity.

The results obtained relate only to men, Fulani pastoralists of the Senegalese Ferlo; the inclusion of women is necessary in the future.

## Figures and Tables

**Table 1 ijerph-20-06999-t001:** Distribution by age and location of the men interviewed (N = 22).

Individual	Age	Village or Camp
H1	39	Village
H2	39	Village
H3	38	Village
H4	38	Camp
H5	42	Village
H6	36	Village
H7	42	Camp
H8	42	Camp
H9	53	Camp
H10	50	Camp
H11	35	Camp
H12	53	Village
H13	53	Camp
H14	51	Village
H15	39	Village
H16	65	Camp
H17	41	Village
H18	50	Village
H19	52	Camp
H20	54	Camp
H21	34	Camp
H22	37	Camp

**Table 2 ijerph-20-06999-t002:** Characteristics of study population (N = 101).

Variables	Categories	N	%	Mean	SD
Age				43.66	17.37
	>15 and ≤ 40 years	45	44.55		
	>40 years	56	55.45		
Occupation	Breeder	54	53.46		
	Shepherd	18	17.82		
	Retired	13	12.87		
	Others (Blacksmiths, merchants, farmers, etc.)	16	15.85		
Are your physical activities always related to your work?	Yes	81	80.20		
	No	14	13.86		
	Sometimes	6	5.94		
Do your physical activities keep you healthy?	Yes	20	19.80		
	No	81	80.20		
Body Mass Index				20.21	3.32
	<18.5 kg/m^2^	37	36.63		
	18.5 kg/m^2^ ≤ BMI < 25 kg/m^2^	54	53.47		
	25 kg/m^2^ ≤ BMI < 30 kg/m^2^	9	8.91		
	>30 kg/m^2^	1	0.99		
Self-rated health	Excellent/very good/Good	73	72.28		
	Fair/Poor	28	27.72		
Total PA according to IPAQ, MET.min/week				6097.4	7264.3
Total vigorous activities (MET.min/week)				4484.3	2646.1
Total moderate activities (MET.min/week)				2669.5	2765.9
Total walking (MET min/week)				1852.5	1852.5
Low (<600 MET)		14	13.86		
Medium (600 ≤ MET < 3000)		17	16.83		
High (≥3000 MET)		70	69.31		
Sitting, min/day				240.66	339.20
Total		101	100.00		

Breeder: farmer who owns his flock and grazes all or part of it, Sheperd: a man who grazes the herd without owning it, an employed shepherd, Retired: those who have stopped working.

**Table 3 ijerph-20-06999-t003:** Pearson correlations between BMI and physical activity components.

	MET Vigorous	MET Moderate	MET Walking	MET Total
Body Mass Index	−0.122 (0.224)	−0.109 (0.276)	−0.303 (0.002 **)	−0.249 (0.012 *)

* *p* < 0.05, ** *p* < 0.01.

**Table 4 ijerph-20-06999-t004:** Regression of self-rated health on age and sitting time (N = 101).

Variables		*p*	OR	CI 95%	
				Inf	Sup
Age (≤40 years)	>40 years	0.134	0.500	0.202	1238
Sitting time (continuous)		0.035 *	0.997	0.995	1000

Note: * *p* < 0.05.

## Data Availability

Datasets (excel sheets and Word documents on interviews) are available on demand.

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
