# Peer review of "Nothing in Excess: Physical Activity, Health, and Life World in Senegalese Fulani Male Pastoralists, a Mixed Method Approach"

_ijerph, 2023, doi:10.3390/ijerph20216999_

Round 1
Reviewer 1 Report
This paper aimed to understand the physical activity of the Senegalese Fulani pastoralists, an ethnic group that has received less attention in previous research. The study utilized both quantitative and qualitative approaches, making it well-designed and original. Most existing studies in this field have focused on industrialized areas, neglecting the diverse health-related perceptions and practices of marginalized groups. Therefore, this research is timely and of significant importance. However, some sections within the paper are still ambiguous and require further clarification. The authors may refer to following comments (both general and in sessions) for further revision.
General comments:
1) This paper focused only on male participants. Although the authors had mentioned it as a limitation in the last part, it is not convincing to generalize the findings to the “life-world” of Fulani. The authors may consider narrowing down the scales of the paper to male in both title and contents so that to make the findings are properly interpreted and contextually situated in the Fulani society.
2) The result session is not well organized. Discussions are mixed with results. Particularly, the narratives are not categorized properly making the reading became very difficult. I suggest the authors re-organize the result session by following themes.
3) This paper lacks an independent discussion session that interactively engages two types of data together to further communicate with existing theories. Particularly, narrative data collected through interviews are not well organized in discussions, with very poor theoretical support. The authors had mentioned body culture theories (e.g., Bernard Andrieu; Marcel Mauss), but those were not introduced adequately nor actively involved in the discussion. I suggest the authors consider introducing them in introductions and go back to these theories by relating them to current challenges in IPAQ assessment in the discussion part based on what you have found.
4) This study primarily utilized interviews, which may induce discrepancies between self-reported data and actual walking and sitting hours. It is important to acknowledge and address this limitation.
Comments to sessions:
Introduction –
(lines 51-74) Theoretical Framework:
This part lacks theoretical support from previous studies. The authors need to add explanations of theories, based on which they develop this study. Also, the first paragraph is very redundant, repeating similar opinions.
(lines 80-121) Context:
The author mentioned, “There is a lack of information in physically active populations from low- and middle-income countries”. This is not true. For instance, Tian et al. (2021) focused on the physical activity of children in pastoralist Maasai society in Sub-Saharan Africa in their paper; Fratkin & Roth (2005) discussed health and mobility issues of pastoralist groups that are under social changes. These studies should be addressed so that to give a comprehensive picture of existing literature on this topic.
(lines 87-89) The authors wrote, “Studying the socio-economic, demographic and health situation of the Fulani population of the Senegalese Ferlo will address this gap in knowledge.” But this study did not focus on socio-economic and demographic issues.
As many readers are not familiar with your study population, explanations of the livelihood of this ethnic group, including age system, types of livestock, labor division or cooperation, roles of men and women in the local economy, distance from the village to camp, etc., is necessary. Moreover, as the targeted population is adult males in this paper, the social roles of adult males should be explained.
(lines 103 -121) The authors listed several social norms that may be related to health and fitness in the world of the life of Fulani people. These aspects are interesting but are not effectively recalled in other sessions, nor being well engaged in discussions. I suggest the authors consider re-organize the narratives in the Results session, linking the narratives of local adult males in different norm categories.
(lines 122-139) Object, Problem and Issue:
Social changes were mentioned, but very general. How adult men’s life has changed under these external forces should be addressed
In the existing studies utilizing the IPAQ-SF, the focus has been on both males and females. However, in this particular study, the authors have chosen to specifically focus on adult males. It is important for the authors to provide an explanation for this choice in order to justify the exclusion of females from the study.
Methods –
(lines 157-160) The author should move basic knowledge of IPAQ into Introduction session. How the data would be calculated should be moved to analysis (lines 160-171). Same to “Body Mass Index”, methods for data analysis should be moved to the sub-session of analysis. “Socio-demographic variables” move to sampling (lines 196-198).
(lines 217-218) “All participants were informed 217 of the voluntary and free nature of their participation.” should be moved to Ethical Concerns. In the section on Ethical Concerns, it is important for the authors to include information about whether they obtained consent from local authorities and provide details about the nature of the consent, including who it was obtained from.
Results –
(line 245) What is the difference of Breeder and Shepherd in table 2? Does “retired” refer to local labor system or more general national wage system? These occupations should be explained.
(lines 249-255) Discussions should be removed from the results section: lines 257-260, 278-282, 352-356, 359-361, 383-385, and 423-440. The authors should distinguish their own findings as "results of this study" and separate them from the discussion, which involves comparisons and references to existing literature.
Conclusion –
Several sentences, such as lines 465-468, are discussions and should be revised by removing them and including them in the Discussion section. The remaining part should be concise and focused.
Possible discrepancies between self-reported data and actual walking and sitting hours should be acknowledged and addressed.
References –
Please recheck the referencing style. “,” is missing between the last name and first name initial in some cases.
Author Response
The authors would like to thank reviewer 1 for his comments and corrections. Below are the authors' responses to these, and in the text of the article, in red, the corrections and clarifications made in response to reviewer 1. Also in the article, in dark yellow, the corrections made in response to reviewer 1 & reviewer 2 or 3.
General comments:
1) This paper focused only on male participants. Although the authors had mentioned it as a limitation in the last part, it is not convincing to generalize the findings to the “life-world” of Fulani. The authors may consider narrowing down the scales of the paper to male in both title and contents so that to make the findings are properly interpreted and contextually situated in the Fulani society.
Authors : Thanks to reviewer 1 for this comment. Indeed, the use of the notion of "lifeworld" here refers to both the cultural (lifestyles and practices) and social (functions, structures, etc.) framework of Fulani pastoralists. It is not a generalization based on the study population (Fulani men), which would indeed be an abusive generalization, but is understood as a socio-cultural framework in which these Fulani men evolve. As reviewer 1 points out, specifying that this applies only to Fulani male pastoralists, regularly in the article as well as in the title that this study has been done :
- Title: Nothing in excess: physical activity, health and life world in Senegalese Fulani male pastoralists
- Line 52 : "Furthermore, we have attempted to define the dimensions of physical activity that the IPAQ-SF captures in this isolated rural environment and, more importantly, what it means to be physically active in this population of Ferlo Fulani men"
- line 135 : "Thus, for both cultural and material reasons, recurrent and relatively intense daily physical activity is characteristic of the Fulani populations of the Ferlo, particularly among men, who are responsible for grazing the herds.
- line 140-143: "In this general context, we proposed to evaluate what the IPAQ-SF was associated with in this population of Senegalese male pastoralists (i.e. health, BMI) by taking into account what this physical activity means for the populations in which it is measured through this scale"
2) The result session is not well organized. Discussions are mixed with results. Particularly, the narratives are not categorized properly making the reading became very difficult. I suggest the authors re-organize the result session by following themes.
Authors: Being accustomed to publishing articles in quantitative journals, we fully understand why Reviewer 1 would make such a justified remark. However, our relatively innovative approach to this mixed-methods subject requires the simultaneous reading of quantitative and qualitative data and their interpretations. On the other hand, in order to facilitate and clarify the reading of this section, we have rearranged the text to make it easier to read and interpret the verbatims.
3) This paper lacks an independent discussion session that interactively engages two types of data together to further communicate with existing theories.
Authors: Indeed, as explained above, the mixed-method approach adopted makes it particularly difficult to separate results from discussion, hence the choice made in this article to allow simultaneous reading of results and their interpretation.
Particularly, narrative data collected through interviews are not well organized in discussions, with very poor theoretical support. The authors had mentioned body culture theories (e.g., Bernard Andrieu; Marcel Mauss), but those were not introduced adequately nor actively involved in the discussion. I suggest the authors consider introducing them in introductions and go back to these theories by relating them to current challenges in IPAQ assessment in the discussion part based on what you have found.
Authors: In the present state of our knowledge, it seems difficult to introduce Andrieu or Mauss as a basis for a critique of the IPAQ evaluation among the Fulani pastoralists of the Ferlo. On the other hand, we have better founded our interpretations by appealing to new references, allowing theories of body culture to shed light on these elements of discussion.
In the theorical framework section :
- Raveneau et Fournier, 2018
- Boltanski, 2008
In the discussion
- Raveneau et Fournier, 2018
- Boltanski, 2008
- Andrieu et al., 2018
- Mauss, 1950
4) This study primarily utilized interviews, which may induce discrepancies between self-reported data and actual walking and sitting hours. It is important to acknowledge and address this limitation.
We thank reviewer 1 for this very pertinent remark. Fortunately, in 2019 our physiologist colleagues measured the number of hours of actual walking on some thirty Fulani male herders. The results (unpublished) showed that declarative hours and effective hours were comparable.
However, to remove any ambiguity, we have included in the limits of the study the fact that a discrepancy could be observed between declarative data and actual PA hours.
- Line 500-502: since a discrepancy can obviously be observed between the declarative measurement of physical activity (IPAQ SF) and its physiological measurement.
Comments to sessions:
Introduction –
(lines 51-74) Theoretical Framework:
This part lacks theoretical support from previous studies. The authors need to add explanations of theories, based on which they develop this study. Also, the first paragraph is very redundant, repeating similar opinions.
Authors: The authors would like to thank reviewer 1 for his editorial comments. We have reworked the text and removed redundancies. In addition, in line with reviewer 1's comment no. 3, the text has been modified to reinforce the theoretical underpinnings of this study.
(lines 80-121) Context:
The author mentioned, “There is a lack of information in physically active populations from low- and middle-income countries”. This is not true. For instance, Tian et al. (2021) focused on the physical activity of children in pastoralist Maasai society in Sub-Saharan Africa in their paper; Fratkin & Roth (2005) discussed health and mobility issues of pastoralist groups that are under social changes. These studies should be addressed so that to give a comprehensive picture of existing literature on this topic.
Authors: Thanks to reviewer 1 for these important references, which we learned a lot from reading. We have therefore specified the existence of these studies, although Tian's study focuses only on children (from an anthropological and cultural point of view) and Fratkin and Ross's study deals with health issues in Kenya and does not directly address the question of IPAQ.
- lines 86-87: Although a few articles focus on physical activity practices in sub-Saharan African rural areas (Tian et al., 2021; Fratkin and Ross, 2005)
(lines 87-89) The authors wrote, “Studying the socio-economic, demographic and health situation of the Fulani population of the Senegalese Ferlo will address this gap in knowledge.” But this study did not focus on socio-economic and demographic issues.
Authors: we apologize to reviewer 1 for the poor wording.The authors actually meant to write: "That's why it seems necessary to describe the socio-economic, demographic and health situation of the Fulani population in Senegal's Ferlo region.".
The text has been changed accordingly in the article (line 88-90)
As many readers are not familiar with your study population, explanations of the livelihood of this ethnic group, including age system, types of livestock, labor division or cooperation, roles of men and women in the local economy, distance from the village to camp, etc., is necessary. Moreover, as the targeted population is adult males in this paper, the social roles of adult males should be explained.
Authors: We have specified some socio-economic data on the study population and the distribution of gender roles (lines 94,95,96 & 100).
(lines 103 -121) The authors listed several social norms that may be related to health and fitness in the world of the life of Fulani people. These aspects are interesting but are not effectively recalled in other sessions, nor being well engaged in discussions. I suggest the authors consider re-organize the narratives in the Results session, linking the narratives of local adult males in different norm categories.
Authors: in fact, these aspects are nevertheless addressed in the discussion when we discuss the constraints of environmental necessity and existence as normative constraints associated with the ndimagu set out in the lines pointed out by the reviewer. But, like the responses to the need to distinguish between results and discussion, it seems difficult to break up the analysis without undermining the coherence of the article.
(lines 122-139) Object, Problem and Issue:
Social changes were mentioned, but very general. How adult men’s life has changed under these external forces should be addressed
Authors: We have clarified these social changes and referred to a study on the subject :
- lines 131-133: "Even if herders are currently diversifying their income (real estate, trade, transport), livestock breeding remains the cornerstone of the Fulani people's way of life [36]"
In the existing studies utilizing the IPAQ-SF, the focus has been on both males and females. However, in this particular study, the authors have chosen to specifically focus on adult males. It is important for the authors to provide an explanation for this choice in order to justify the exclusion of females from the study.
Authors: There are several empirical and theoretical reasons for prioritizing Fulani male pastoralists. On the one hand, we focused on individuals who a priori had the most continuous physical activity, given our subject. In addition, our physio colleagues Bourdier et al (2021) had already studied this population of Fulani male herders in 2019. Furthermore, the reasons also relate to feasibility during laboratory missions. The data were gendered because physical activities are gendered, and studying men and women at the same time would not only have been impossible in the field during missions, but inconsistent given the gendered partition of this society in physical activities linked to the demands and necessities of their lives, their social organization, and above all the existential constraints of vital conditions.
Methods –
(lines 157-160) The author should move basic knowledge of IPAQ into Introduction session. How the data would be calculated should be moved to analysis (lines 160-171). Same to “Body Mass Index”, methods for data analysis should be moved to the sub-session of analysis. “Socio-demographic variables” move to sampling (lines 196-198).
Authors: We would like to thank reviewer 1 for his suggestions: we have moved lines 157-160 to the introduction (now lines 35-38 in dark yellow since reviewer 3 made the same remark). However, as regards the organization of the categories linked to each variable, we have based ourselves on the articles previously published in the IJERPH journal and we therefore wish to retain this organization, if it suits the editors.
(lines 217-218) “All participants were informed 217 of the voluntary and free nature of their participation.” should be moved to Ethical Concerns. In the section on Ethical Concerns, it is important for the authors to include information about whether they obtained consent from local authorities and provide details about the nature of the consent, including who it was obtained from.
Authors: The study obtained the authorization of the national health research ethics committee indicated below in the article, which included the consent form sent to participants.
Results –
(line 245) What is the difference of Breeder and Shepherd in table 2? Does “retired” refer to local labor system or more general national wage system? These occupations should be explained.
Authors: The terms used are:
- Shepherd is a man who grazes the herd without owning it, an employed shepherd; whereas
- Breeder is a farmer who owns his flock and grazes all or part of it.
- Retired" refers to the local work system, as there is no general pension system for all workers in Senegal, apart from civil servants and special schemes for certain companies. In the table, therefore, this term refers to those who have stopped working.
These elements have been specified on lines 252-254.
(lines 249-255) Discussions should be removed from the results section: lines 257-260, 278-282, 352-356, 359-361, 383-385, and 423-440. The authors should distinguish their own findings as "results of this study" and separate them from the discussion, which involves comparisons and references to existing literature.
Authors: As stated on several occasions, and in particular in general remark no. 2, the dissociation between results and discussion seems to us to be of little relevance in the context of our study.
Conclusion –
Several sentences, such as lines 465-468, are discussions and should be revised by removing them and including them in the Discussion section. The remaining part should be concise and focused.
Authors: thanks to the reviewer for this remark, the sentence has been moved to the end of the previous section (lines 459-461). tWe've tried to focus as much as possible, given the study we've carried out, and we've made this passage more concise (lines 478-479).
Possible discrepancies between self-reported data and actual walking and sitting hours should be acknowledged and addressed.
Authors: We thank reviewer 1 for this very pertinent remark. To remove any ambiguity, we have included in the limits of the study the fact that a discrepancy could be observed between declarative data and actual Physical activities hours.
- Line 491-493: since a discrepancy can obviously be observed between the declarative measurement of physical activity (IPAQ SF) and its physiological measurement.
References –
Please recheck the referencing style. “,” is missing between the last name and first name initial in some cases.
Authors: The bibliography has been corrected to conform to editorial standards.

Reviewer 2 Report
I like this paper very much. I like it not for it's strength of research but for the topic and the importance of it's social commentary on perception of importance of exercise. I read every word and thought about the meaningfulness of this subject matter. I think the paper's work is not in the hard data that was found but on the importance what a social environment does in affecting behavior. For that, I think the paper should be published.
However, there are some badly needed corrections in the results sections.
Not all the data are reported in the tables, for example widowhood is not reported. p. 7. In that same paragraph data are reported from other populations that are not available in tables or references. Tables are mis-labed. On page 10, fourth full paragraph, Table 3 supposedly shows information on sitting from Table 3, which is really in Table 2. I recommend the authors critically review what they are saying in the results and match to the right tables.
Also, numerous errors in reference form. I will list the reference number that has errors that need to be corrected. The format used in the following references do not match similar format in other references.
Reference: 3, 4, 7, 9, 10, 12, 13, 17, 18, 19, 20, 22, 24, 25, 26, 34,35, 37, 39,40, 44, 46, 47,51.
Author Response
The authors would like to thank reviewer 2 for his comments and corrections. Below are the authors' responses to these, and in the text of the article, in green, the corrections and clarifications made in response to reviewer 2. Also in the article, in dark yellow, the corrections made in response to reviewer 2 & reviewer 1 or 3.
I like this paper very much. I like it not for it's strength of research but for the topic and the importance of it's social commentary on perception of importance of exercise. I read every word and thought about the meaningfulness of this subject matter. I think the paper's work is not in the hard data that was found but on the importance what a social environment does in affecting behavior. For that, I think the paper should be published.
However, there are some badly needed corrections in the results sections.
Not all the data are reported in the tables, for example widowhood is not reported. p. 7.
Authors: We have explained this point in lines 258-259 : in fact, the practice of polygamy results in virtually no widowhood for men.
In that same paragraph data are reported from other populations that are not available in tables or references.
Authors: The data are compared to a population of Nigeria the reference is quoted: [6] line 259. Thanks to the reviewer.
Tables are mis-labed. On page 10, fourth full paragraph, Table 3 supposedly shows information on sitting from Table 3, which is really in Table 2. I recommend the authors critically review what they are saying in the results and match to the right tables.
Authors: Thanks to reviewer 2, error corrected on line 406
Also, numerous errors in reference form. I will list the reference number that has errors that need to be corrected. The format used in the following references do not match similar format in other references.
Reference: 3, 4, 7, 9, 10, 12, 13, 17, 18, 19, 20, 22, 24, 25, 26, 34,35, 37, 39,40, 44, 46, 47,51.
Authors: Thank you very much for these remarks. The bibliography has been corrected to conform to editorial standards.

Reviewer 3 Report
Greetings
Thanks for giving me the opportunity of reviewing the valuable manuscript entitle” Nothing in excess: physical activity, health and life world in Senegalese Fulani pastoralists, a mixed method approach”
The following points are suggested:
Writing revisions are required in all parts of the manuscript.
Title:
Abstract:
Line 6: the word” test” is better to substitute by evaluate.
Line 9: the measurement categories of the questionnaire need revision.
Line 11: The word: study” seems not necessary.
Line 23: Conclusion is not the reflection of the study aims.
Line 23: The phrase “Time spent walking” needs rewording
Introduction:
Line 33: respectively must move to after the word “death”
Line 40-41 need rewording
Line 46: needs rewording
In introduction and in the final paragraph evidence must be presented that IPAQ is differently applicable and validated in various cultures.
Theoretical framework:
Line 52- 53 need rewording.
Several sentences are available in the 1st paragraph without referencing,
Context
Some information is not related to aims of the study and seems unnecessary
Methods
Inclusion and exclusion criteria must be mentioned.
I could not find the number of population (not participants) of the study.
It must be included that whether the participants language was English? If not. How the translated version of questionnaire was validated?
Lines 157-160 needs more clarifications by rewording.
Line 152: number of days…. You mean per week?
Line 328: needs rewording
Results:
Table 2 : it is recommended that age ranges of each category be presents: eg. >20≤40
Conclusion:
Question:
IPAQ includes 5 categories and only one category (as sport or exercise) is related to sport which many not be common in this population of the study. I think this section can be ignored in assessing the daily physical activity and interpretation about the validity of this questionnaire in this special community seems need revision. Or if possible please present your reason.
The other point is that high intensity physical activity is not related to sport activities and is possible in all our daily activities so the related questions need interpretations and must be accompanied with related examples to be understandable for participants. In this part also I think explanations is necessary ; why in several parts it has been mentioned as unapplicable for the target community.
quality of English writing must be modified in several parts
Author Response
The authors would like to thank reviewer 3 for his comments and corrections. Below are the authors' responses to these, and in the text of the article, in purple, the corrections and clarifications made in response to reviewer 3. Also in the article, in dark yellow, the corrections made in response to reviewer 3 & reviewer 2 or 3.
Greetings
Thanks for giving me the opportunity of reviewing the valuable manuscript entitle” Nothing in excess: physical activity, health and life world in Senegalese Fulani pastoralists, a mixed method approach”
The following points are suggested:
Writing revisions are required in all parts of the manuscript.
Title:
Abstract:
Line 6: the word” test” is better to substitute by evaluate.
Authors: thank you for this remark, correction line 6 "evaluate".
Line 9: the measurement categories of the questionnaire need revision.
Authors: thank you for this remark, correction made line 9.
Line 11: The word: study” seems not necessary.
Authors: thank you for this remark, correction made line 11.
Line 23: Conclusion is not the reflection of the study aims.
Authors: Thank you, the conclusion has been modified to better match the study's objectives.
- line 23-26: "the mixed method approach developed in our article has shown that the IPAQ SF is not a valid measure of physical activity in the population of Fulani male herders from the Ferlo region, given that unproductive energy expenditure is incompatible with the Fulani way of life, which condemns excess and immoderation"
Line 23: The phrase “Time spent walking” needs rewording
Authors: Thank you, tthe term has been deleted in the rewording.
Introduction:
Line 33: respectively must move to after the word “death”
Authors: thank you for this remark, correction made line 33.
Line 40-41 need rewording
Authors: Thank you, the introduction has been modified lines 43-47
- "It is essential to validate tools for measuring daily physical activity in low- and middle-income countries, given the considerable increase in the prevalence of chronic diseases in these countries [11], in addition to the high mortality rate linked to non-communicable diseases [12]."
Line 46: needs rewording
Authors: Thank you, the conclusion has been modified lines 49-51
- "Furthermore, we have attempted to define the dimensions of physical activity that the IPAQ-SF captures in this isolated rural environment and, more importantly, what it means to be physically active in this population of Ferlo Fulani men."
In introduction and in the final paragraph evidence must be presented that IPAQ is differently applicable and validated in various cultures.
Authors: This was also requested by reviewer 1 and has been added on lines 36-39.
- "The IPAQ-SF has specifications for respondents aged 15 and over, predictive validity, concurrent validity, convergent validity, criterion validity and discriminant validity [6]. These are appropriate research tools, and a good test-retest is an indication of reliability [5,7]."
If reviewer 3 deems it necessary, we can add the following reference:
Honado, Sedonoude Aristide, Batcho Sèbiyo Charles, Roy Jean-Sébastien, and Université Laval. Faculty of Medicine. 2019. "Adaptation Et Validation Du Questionnaire International De L'activité Physique (Ipaq) Chez Les Personnes Saines Et Les Survivants D'un Accident Vasculaire Cérébral Au Bénin." Dissertation, Laval University.
Theoretical framework:
Line 52- 53 need rewording.
Authors: Thank you, the sentence has been deleted in response to reviewer's 1 remark.
Several sentences are available in the 1st paragraph without referencing
Authors: Corrections made in response to reviewer 1 & 3 (in dark yellow in the article)
Context
Some information is not related to aims of the study and seems unnecessary
Authors: Although they sometimes seem a little far removed from the specific objective of this article, the authors feel that information on the context in which the Fulani herders of the Ferlo evolve gives a better grasp of the issues at stake in their world. Moreover, reviewer n°1 asks for more information on the living conditions and lifestyles of this population. However, if reviewer 3 wishes to specifically delete certain passages, the authors are of course willing to do so.
Methods
Inclusion and exclusion criteria must be mentioned.
Authors: Men aged 15 or over living in the commune of Tessekere and not suffering from any disability were included in this study.
- lines 154-155 "... (aged 15 or over, living in the commune of Téssékéré and not physically disabled) "
I could not find the number of population (not participants) of the study.
Authors: This is shown on line 159 for the quantitative study and line 220 for the qualitative study.
It must be included that whether the participants language was English? If not. How the translated version of questionnaire was validated?
Authors: Thank you for your request for clarification. We have responded by adding the items lines 161-164: "The language of the interviewers and respondents was Fulani. The questionnaire had previously been translated from French into Fulani using the parallel inverted translation method (Vallerand, 1989)."
Lines 157-160 needs more clarifications by rewording.
Authors: The wording has been revised for greater clarity and this sentence has been moved to the introduction at the request of reviewers 1 and 3 (in dark yellow in the introduction section)
Line 152: number of days…. You mean per week?
Authors: Thank you for noting this error, it has been corrected line 163 : "Recruitment took place every day for the last two weeks of May 2021."
Line 328: needs rewording
Authors: Thank you, the wording has been changed to clarify lines 337-339 : " Pastoral activities are carried out in difficult and precarious living conditions, so all the physical effort and expenditure of energy, as well as the resulting fatigue, can only be justified by work. "
Results:
Table 2 : it is recommended that age ranges of each category be presents: eg. >20≤40
Authors: correction done in table 2
Conclusion:
Question:
IPAQ includes 5 categories and only one category (as sport or exercise) is related to sport which many not be common in this population of the study. I think this section can be ignored in assessing the daily physical activity and interpretation about the validity of this questionnaire in this special community seems need revision. Or if possible please present your reason.
Authors: We would like to thank reviewer 3 for his question. However, in this study the authors used the IPAQ short form, which only includes 3 categories of physical activity: vigorous, moderate and walking. On the contrary, the IPAQ long form includes a specific dimension of activity linked to leisure, which led us not to choose this scale to study physical activity in Fulani men from the Senegalese Ferlo.
The other point is that high intensity physical activity is not related to sport activities and is possible in all our daily activities so the related questions need interpretations and must be accompanied with related examples to be understandable for participants. In this part also I think explanations is necessary ; why in several parts it has been mentioned as unapplicable for the target community.
Authors: Thank you for your pertinent comment. In the article we added examples to the questionnaire so that people could see which activities were considered vigorous and moderate. This has been done during the parallel-reverse translation process. The examples of intense physical activity given to the participants were adapted to the context and were as follows: carrying heavy loads, lifting inner tubes full of water, running... all of which greatly increase breathing. Similarly, for moderate activities, the examples were as follows: carrying light loads, sweeping a broom, walking quickly, all of which increased the breathing rate slightly

Round 2
Reviewer 3 Report
Line 46: another sentence(preferably with reference) must be added that increasing communicable disease can be related to decreasing physical activity.
Minor English revision is required
Author Response
Thanks to reviewer 3 for this remark. Line 46, we have included the following sentece :
"Moreover, the relationship between the increase in chronic diseases and the decrease in physical activity [13] makes it all the more necessary to address this gap in knowledge."
English language has been edited.